# A Combined In-Mold Decoration and Microcellular Injection Molding Method for Preparing Foamed Products with Improved Surface Appearance

**DOI:** 10.3390/polym11050778

**Published:** 2019-05-01

**Authors:** Wei Guo, Qing Yang, Huajie Mao, Zhenghua Meng, Lin Hua, Bo He

**Affiliations:** 1School of Automotive Engineering, Wuhan University of Technology, Wuhan 430070, China; whutgw@126.com (W.G.); yqwhut@163.com (Q.Y.); hualin@whut.edu.cn (L.H.); 2Hubei Key Laboratory of Advanced Technology for Automotive Components, Wuhan University of Technology, Wuhan 430070, China; 3Hubei Collaborative Innovation Center for Automotive Components Technology, Wuhan 430070, China; 4School of Materials Science and Engineering, Wuhan University of Technology, Wuhan 430070, China; hbwhut@163.com

**Keywords:** in-mold decoration injection molding, microcellular injection molding, surface quality, mechanical properties, warpage

## Abstract

A combined in-mold decoration and microcellular injection molding (IMD/MIM) method by integrating in-mold decoration injection molding (IMD) with microcellular injection molding (MIM) was proposed in this paper. To verify the effectiveness of the IMD/MIM method, comparisons of in-mold decoration injection molding (IMD), conventional injection molding (CIM), IMD/MIM and microcellular injection molding (MIM) simulations and experiments were performed. The results show that compared with MIM, the film flattens the bubbles that have not been cooled and turned to the surface, thus improving the surface quality of the parts. The existence of the film results in an asymmetrical temperature distribution along the thickness of the sample, and the higher temperature on the film side leads the cell to move toward it, thus obtaining a cell-offset part. However, the mechanical properties of the IMD/MIM splines are degraded due to the presence of cells, while specific mechanical properties similar to their solid counterparts are maintained. Besides, the existence of the film reduces the heat transfer coefficient of the film side so that the sides of the part are cooled asymmetrically, causing warpage.

## 1. Introduction

Microcellular injection molding (MIM) technology originated from the idea of Nam Suh [1]. Compared with conventional injection molding (CIM), it has some advantages including weight reduction, cost saving, and excellent dimensional stability [2,3,4,5]. Despite these advantages, however, surface defects-such as gas flow marks and lack of smoothness-still remain one of the main drawbacks of the MIM parts, which limit MIM’s application to a large extent [6].

In MIM, the polymer melt and supercritical fluid (SCF) are mixed to form the polymer/SCF single-phase solution in the barrel; the polymer/SCF single-phase solution is then injected into the mold cavity under the action of the screw. The large pressure drop causes thermodynamic instability, which provides conditions for bubble nucleation and growth. As the bubble grows, it is stretched, compressed, and even broken under the action of the shear flow. Due to the fountain flow behavior at the front of the melt, the bubble turns over to the cavity surface and solidifies under the cooling of the mold, leaving a so-called bubble mark on the surface of the part, which greatly affects the surface quality [7]. There have been many attempts to eliminate bubble marks to improve the surface quality. For example, rapid thermal cycle molding (RTCM) technology was developed to eliminate surface defects of the microcellular injection molded parts [8,9,10,11,12]. By increasing the temperature of the mold cavity surface, the fluidity of the melt can be increased, thereby increasing the filling ability of the melt to dissolve the bubble marks back into the melt [13]. Besides, the high-temperature mold flattens the bubble marks which have not been cooled and formed to be turned over to the mold cavity surface, and the microcellular foamed parts with a good surface quality are obtained.

Integrating co-injection with microcellular injection molding can also solve the surface appearance issue [14,15,16,17]. Microcellular co-injection molding injects the surface layer material before injecting the core material to form a sandwich structure in which the surface layer is not foamed, and the core layer is foamed. Since the surface layer is solid, the parts with good surface appearance are obtained.

It is also possible to use a combination of gas counter pressure (GCP) and microcellular injection molding to eliminate flow marks to improve surface quality [18,19,20]. GCP is the way to apply high pressure to the mold cavity before melt injection and to maintain a constant back pressure during the melt injection. Constant back pressure can suppress the nucleation and growth of the bubbles, and the bubbles will start to grow after the back pressure is released. At this time, the surface layer has solidified, thus improving the surface quality by reducing the bubble marks on the surface.

There have also been efforts to improve surface quality by gas-assisted microcellular injection molding [21]. Hou at el. found that the cells generated during the melt filling process could be dissolved back into the polymer melt through the high-pressure assisted gas from the GAMIM, thus improving the part’s surface appearance by eliminating the sliver marks. Although these methods do have improved surface quality, they all require additional equipment, which means improvements of the experimental equipment, thus greatly increasing the cost of the experiment.

In addition, since the conventional injection molded parts need to be decorated with some patterns and logos after the injection to improve the aesthetics and surface properties, the conventional surface decoration techniques such as spraying and plating have many drawbacks, such as high defect rate, wear resistance, long production cycle, and severe environmental pollution. To solve these problems, the in-mold decoration (IMD) technique has been proposed. The IMD method integrates injection molding and decoration, thus eliminating the secondary processing of conventional injection molding. Compared with conventional injection molding, it not only reduces labor costs, but also improves work efficiency, and the injection molded parts have better stability and durability [22,23,24,25].

To solve the shortcomings of microcellular injection molding, a combined in-mold decoration and microcellular injection molding (IMD/MIM) method was proposed in this paper. The IMD, CIM, IMD/MIM and MIM simulations and experiments were conducted to verify the effectiveness of the IMD/MIM method. The mechanical properties, forming defects and cellular structure of the samples were characterized to compare the IMD, CIM, IMD/MIM and MIM.

## 2. Combined In-Mold Decoration and Microcellular Injection Molding

The schematic diagram of the combined in-mold decoration and microcellular injection molding (IMD/MIM) method is illustrated in Figure 1. The PET film is attached to the mold cavity before injection, the polymer melt and supercritical fluid (SCF) are mixed under the stirring of the screw to form the polymer/SCF single-phase solution in the barrel for injection, as can be seen in Figure 1a. In Figure 1b, the polymer/SCF single-phase solution is injected into the cavity under the action of the screw. The large pressure drop causes thermodynamic instability, which provides conditions for bubble nucleation and growth. Due to the presence of the film, the heat transfer coefficient on the film side is reduced, resulting in an asymmetrical temperature distribution along the thickness of the sample, thus the flow is asymmetrical. In Figure 1c, the melt filling is completed, the bubbles continue to grow until the melt solidifies. At the end of the filling, the melt temperature on the film side is higher than that on the non-film side, and the viscosity is lower than that of the non-film side. Finally, the mold is opened, and the flexural sample is taken out as depicted in Figure 1d. The bubble is shifted toward the film side due to the higher temperature on the film side, and some cells undergo significant deformation under shear, extension, and compression of the shear flow, which is caused by the fountain flow behavior at the melt front.

## 3. Experimental Setup

### 3.1. Numerical Simulation

#### 3.1.1. Finite Element Model

To correspond to the experiment, a 4-cavity was used, and the flexural sample (80 × 10 × 4 mm) was selected as the research object. The decorative film (PET) with the thickness of 0.2 mm was attached to the mold cavity surface, and the cooling system and runner system were established as required in Moldflow as shown in Figure 2. The main processing parameters were set as follows: melt temperature 220 °C, mold temperature 50 °C, coolant temperature 25 °C, foaming agent’s content 0.5%; the bubble nucleation model used the fitted classical nucleation model due to the asymmetric mold cavity temperature distribution which affects bubble nucleation.

#### 3.1.2. Experimental Apparatus

A PVT testing machine (PVT-6000, Gotech, Beijing, China) was used to measure the PVT performance; the method adopted was the isostatic cooling method, the pressure was set to 30, 60, 90 and 120 MPa, the temperature was heated to 200 °C, and the cooling rate was 5 °C/min.

Viscosity was measured using the capillary rheometer (CR-6000, Gotech, Beijing, China); the experimental method used was to keep the temperature constant, pressurize, and then viscosity was measured at different shear rates, and the experiment was repeated at different temperatures (180, 200, 220 and 240 °C).

Cellular structure and surface topography were observed using a JSM-IT300 (JEOL Ltd., Tokyo, Japan) scanning electron microscope (SEM). The samples were ruptured after immersion in liquid nitrogen for 3 h before SEM observation. Then, platinum was sprayed on the fracture surface to conduct electricity.

A 3D optical profiler (ST400, NANOVEA Corporation, Irvine, CA, USA) was used to analyze the surface profile of the samples. Warpage was measured with a 3D laser scan (Handy scan 700, Creaform, Wuxi, China).

The tensile, flexural and impact tests were performed to characterize the mechanical properties of the samples. Tensile and flexural tests were performed using an electromechanical universal testing machine (CMT6104, MTS Systems Corp, Eden Prairie, MN, USA). The impact test was performed using an impact tester (XJUD-5.5, Chengde Jinjian Testing Instrument Co., Ltd., Chengde, China). The tensile test method was ISO 527-1:1993 with a crosshead speed of 50 mm/min. The flexural test method was ISO 178:2001 with a speed of 2 mm/min. The impact strength was measured according to ISO 180:2000. To avoid errors caused by the contingency of the test, five samples were tested under each test condition, and their average values were taken as the results.

#### 3.1.3. Materials

The polymer material and physical foaming agent were modified PP, grade AIP-1927 (supplied by Kingfa Sci & Tech Co., Ltd., Guangzhou, China) and N_2_ (supplied by Wuhan Xiangyun Industry and Trade Co., Ltd., Wuhan, China), respectively, and PET film with the thickness of 0.2 mm was used as the decorative film. Their properties were depicted in Table 1.

To perform simulation analysis more accurately, the PVT performance and viscosity property of the modified PP were tested, thus obtaining the PVT and viscosity curves as shown in Figure 3a and Figure 4a, respectively. Then, based on the Cross-WLF model and the modified dual-domain Tait equation of state, the PVT and viscosity curves were fitted using the combination of the Mc Quaid method and the general global optimization method (LM-UGO); the fitted PVT and viscosity curves were obtained as illustrated in Figure 3b and Figure 4b, respectively. The fitted PVT and viscosity characteristic parameters were imported into Moldflow for simulation [26].

### 3.2. Experimental Design

The IMD, CIM, IMD/MIM and MIM experiments were carried out using an injection molding machine (HDX50, Ningbo Haida Plastic Machinery Co., Ltd., Ningbo, China) and high-pressure compressor (GBL-200/350, Beijing Zhongtuo Machinery Group Co., Ltd., Beijing, China). The processing parameters used in all experiments are as follows: injection speed 60 g/s, injection pressure 70 MPa, melt temperature 220 °C, mold temperature 50 °C, coolant temperature 25 °C, back pressure 10 MPa, cooling time 25 s, foaming agent’s content 0.5%, gas dosing pressure 17.5 MPa, gas dosing time 3 s. A holding pressure of 35 MPa was applied in the CIM and IMD experiments for a holding time of 12 s.

### 3.3. Characterizations

Three flexural samples on the same cavity were selected as the observation object, as shown in Figure 5. The vertical section (10 × 4 mm), which represents the cross-section of the sample perpendicular to the melt flow direction, was observed to assess the cell structure and distribution. The parallel section (10 × 4 mm), which represents the cross-section of the sample parallel to the melt flow direction, was examined to obtain the orientation and deformation of the cells. The cell diameter and cell number were measured by Image Pro Plus 6.0 based on SEM micrographs. The cell average radius (*R*) was calculated using Equation (1):(1)$$R=\frac{\sum_{i=1}^{n}d_{i}}{2n}$$where *d_i_* indicates the diameter of the *i*_th_ cell within the designated area; *n* represents the number of cells in the given SEM micrograph. The cell density (*N*) was calculated by using Equation (2):(2)$$N={\left(\frac{nM^{2}}{A}\right)}^{\frac{3}{2}}$$where *M* represents the magnification of the SEM micrograph; *A* represents the area of the SEM micrograph.

The ratio of the length to diameter (*c*) was used to characterize the cell deformation and the tilt angle (θ) to characterize the cell orientation. Similarly, the ratio of the length and tilt angle were obtained by Image Pro Plus 6.0. As can be seen from Figure 5, the ratio of length to diameter (*c*) could be calculated as Equation (3):(3)$$c=\frac{L}{B}$$where *L* represents the major axis length of the cell, and *B* represents the minor axis length of the cell. The tilt angle (θ) was defined as the angle between the major axis of the cell and the melt flow direction (sharp angle), which is shown in Figure 6 [27].

The rectangular sample (10 mm × 10 mm) which is located at the center of the flexural sample along the melt flow direction was used for observing surface appearance; the observation surface is depicted in Figure 5. The arithmetic mean roughness (*Ra*) and root mean square roughness (*Rq*) were used to characterize the surface roughness. Both *Ra* and *Rq* were calculated from the 3D optical profiler with a precision of 0.01 μm. Five samples were analyzed for each condition and the average surface roughness obtained were taken as the results.

## 4. Results and Discussion

### 4.1. Mechanical Properties

#### 4.1.1. Tensile Properties

The tensile stress-strain curves of the IMD, CIM, IMD/MIM and MIM samples are given in Figure 7a. As can be seen from Figure 7a, the tensile strength of foamed samples is lower than that of their solid counterparts. This is attributed to the presence of cells, which reduces the bearing area, and some larger cells also cause stress concentration, thus reducing the tensile strength of the spline. The decrease in tensile strength of CIM sample is approximately 0.9%, which is almost negligible when compared with that of IMD sample. It is no exception that the tensile strength of the IMD/MIM sample is only a little higher than that of MIM sample. It can be inferred that the presence of the film has no significant effect on the tensile strength. The specific tensile stress-strain curves of the IMD, CIM, IMD/MIM and MIM samples are obtained by taking the weight reduction into account, as shown in Figure 7b. It can be observed that the specific tensile strength of the foamed samples is a little lower than that of the solid ones, which shows that the foamed samples maintain specific tensile strength similar to that of their solid counterparts. The quantitative comparison of the specific tensile properties such as specific tensile strength, strain-at-break, specific Young’s modulus, and specific toughness of the four kinds of samples are given in Figure 8. The density of the foamed specimens of IMD/MIM and MIM is 13.75% and 15.47% less than that of their solid counterparts, respectively. It is found that the specific tensile strength and specific Young’s modulus of the foamed samples are a litter lower than those of their solid counterparts, which means foamed samples can maintain the same specific tensile properties as those of the solid ones.

#### 4.1.2. Flexural Properties

Figure 9a gives the flexural stress-strain curves of the IMD, CIM, IMD/MIM and MIM samples. As shown in Figure 9a, the foamed samples have a higher flexural strength and elastic modulus than those of their solid counterparts, which indicates that the flexural strength and stiffness of foamed samples are reduced. This is attributed to the presence of cells, which reduces the material on the forced cross-section, thereby decreasing the rigidity and resulting in a reduction in the flexural strength of the spline. It can be seen from Figure 9a that the presence of the film has no significant effect on the flexural strength. The specific flexural stress-strain curves of the IMD, CIM, IMD/MIM and MIM samples are given in Figure 9b. It can be observed that the specific flexural strength of the foamed splines is a little lower than that of the solid ones, which shows that the foamed samples can maintain specific flexural strength similar to that of their solid counterparts. The quantitative comparison of the specific flexural properties (specific flexural strength) of the four kinds of samples is given in Figure 8. As shown in Figure 8, the specific flexural strength of the foamed samples is a little lower than that of the solid ones, which is almost negligible. This means that the specific flexural strength of the foamed samples does not decrease significantly due to the presence of the cells.

#### 4.1.3. Impact Properties

The quantitative comparison of the flexural properties (flexural strength and specific flexural strength) of the IMD, CIM, IMD/MIM and MIM samples is shown in Figure 8. It can be seen that the impact strength of the foamed samples is lower than that of their solid counterparts. The presence of cells helps to passivate the crack tip and effectively prevent further crack propagation. Besides, the deformation of the cells can absorb part of the energy, thereby improving the impact strength. However, uneven cell distribution causes local large cells to become stress concentration points, thereby reducing impact toughness. The combined effect of the two factors results in a decrease in the impact strength. However, it can be seen from Figure 8 that the specific impact strength of the foamed samples is only a little lower than that of their solid counterparts, which means that the foamed samples can maintain the same specific impact properties as those of the solid ones. Similarly, the presence of the film has no significant effect on the impact strength, as shown in Figure 8.

### 4.2. Formation Defects

#### 4.2.1. Surface Topography

The surface topography, surface profile, experimental volume shrinkage and simulated volume shrinkage of the IMD, CIM, IMD/MIM and MIM samples are shown in Figure 10. It can be seen that the MIM sample has a rough surface with many silver streaks and cracked bubbles, while the surface of the IMD/MIM sample is as smooth as those of the IMD and CIM samples, with a small amount of burst bubbles. Due to the fountain flow behavior at the front of the melt, the grown bubbles turn to the sides, reach the mold surface and deform or even rupture under the shearing and stretching of the shear flow. Under the action of the mold, the bubbles are flattened on the surface to form bubble marks, which greatly reduces the surface quality of the part. However, the surface quality of IMD/MIM sample is almost as good as that of CIM sample, with few bubble marks on the surface. The presence of the film, which reduces the heat transfer coefficient on the film side, results in a higher melt temperature on the film side and a lower cooling rate. When the bubbles are turned over to the surface of the mold, they are flattened by the film before being completely cooled. From the surface profile in Figure 10, we can find that the solid samples have shrunk while the foamed splines have not, which is consistent with the results of simulation and experiment. Due to the growth of the cells, the voids created by the shrinkage of the melt are supplemented, allowing the melt to fill the entire cavity and reducing the shrinkage of the samples.

The surface roughness of the IMD, CIM, IMD/MIM and MIM samples is shown in Figure 11. It can be found that the surface roughness of IMD sample is smaller than that of CIM sample, and the roughness of IMD/MIM sample is smaller than that of MIM sample, which means that the presence of the film reduces the surface roughness, so that the sample has a better surface quality. Compared with the CIM sample, the MIM sample has much larger surface roughness, which is attributed to the presence of many bubble marks on the surface of MIM parts, resulting in the poor surface quality. However, the surface roughness of IMD/MIM is only a little smaller than that of IMD samples, which is almost negligible. The presence of the film compensates for the defect of the surface quality of the foamed samples. The result is consistent with that of the surface topography, so it can be concluded that the IMD/MIM method does improve surface quality of the foamed samples, which plays a significant role in the promotion and application of microcellular injection molding technology.

#### 4.2.2. Warpage

The warpage of the IMD, CIM, IMD/MIM, and MIM samples is shown in Figure 12. It is observed that the warpage value of the IMD sample is 2.961 mm, which is much greater than that (0.7144 mm) of CIM sample, which is consistent with the experimental result. The existence of the film reduces the heat transfer coefficient on the film side, so that the cooling rate of the melt on the film side is lower than that of the non-film side, resulting in an asymmetrical temperature on both sides of the spline, and the temperature difference between the two sides at the end of the melt filling (35.13 °C) is much greater than that (5.92 °C) of CIM sample, as shown in Figure 12. Therefore, the two sides are not uniformly shrunk, and the sample is bent toward the high-temperature side, that is, the film side. Besides, the temperature at each point on the part surface is also different, resulting in different shrinkage rates everywhere, which also increases the warpage. The warpage of the IMD/MIM sample (1.898 mm) is much larger than that of the MIM sample (0.5887 mm), but the warpage is reduced compared with that of the IMD sample, which is consistent with the experimental result. The growth of the cells facilitates the replenishment of the voids caused by the shrinkage of the melt, so that the melt fills the entire cavity, thereby reducing the warpage to some extent.

### 4.3. Cellular Structure

Since the cells in the vertical section are hardly deformed, we can divide the vertical section of the MIM and IMD/MIM samples into three layers, namely, the transition layer A, transition layer B and core layer C based on the cell size distribution, as shown in Figure 13. For the cells in the parallel section, they are stretched and compressed under the action of the shear flow, resulting in deformation. We no longer stratify according to the cell size distribution, but determine the boundary between the transition layer and the core layer according to the degree of cell deformation.

#### 4.3.1. Cellular Structure of Vertical Section

The simulation results and SEM micrographs of the vertical section of MIM and IMD/MIM samples are given in Figure 14. It can be seen that whether it is the MIM or IMD/MIM sample, the cell size of the core layer is larger than that of the transition layer, which is consistent with the cell size distribution in Figure 14a,b. Combined with the temperature curves in Figure 14e,f, it can be found that the temperature distribution along the thickness direction of the sample is not uniform, and the temperature of the core layer is higher than that of the transition layer, which provides a longer time for cell growth. Therefore, there is such a phenomenon that the core layer has large cells and the transition layer cells are small. As can be seen from Figure 14a,b, the cell radius and tensile modulus change in the opposite direction in the thickness direction, and the maximum cell radius corresponds to the minimum tensile modulus, which illustrates that the cell size affects the tensile modulus. The cell density of the core layer is less than that of the transition layer. In combination with the viscosity curves of Figure 14e,f, higher temperature of core layer provides longer growth times for cells, while lower viscosity results in lower melt strength, thus reducing cell growth resistance. Therefore, the cells are merged and collapsed more, resulting in the cell density of the core layer being lower than that of the transition layer.

As can be seen from Figure 15, it is not difficult to find that the cells of the two transition layers of the MIM sample are similar in size and density. However, the cell of transition layer B (with film side) of the IMD/MIM sample is larger than that of the transition layer A, while the cell density on both sides is not much different. The presence of the film reduces the heat transfer coefficient on the film side, so cells have a longer time to grow due to the higher temperature of the film side. At the same time, due to the higher temperature, the solubility of the gas in the melt is lowered, the thermodynamic instability of the gas is enhanced, thus increasing the nucleation rate of cells. However, the number of cell nucleation is primarily determined by the thermodynamic instability induced by the large pressure drop that occurs when the polymer/SCF single-phase solution is injected into the mold cavity, so that there is no significant difference in the cell density of the two transition layers.

From Figure 14c,d, we can see that the largest cell of the MIM sample is on the center line, while the largest cell of the IMD/MIM sample is offset from the center line toward the film side. Due to the presence of the film, the temperature on the film side is higher than that on the non-film side. The maximum temperature no longer appears on the center line of the sample thickness, but is shifted to a certain distance toward the film side, so the largest cell of the IMD/MIM is biased toward the film side. The distance from the left edge of the MIM core layer to the left surface is 1.41 mm, and the distance from the right edge to the right surface is 1.39 mm, while the distance from the left edge of the IMD/MIM core layer to the left surface is approximately 1.49 mm and the distance from the right edge to the right surface is approximately 1.31 mm. It can be clearly seen that the core layer of the MIM is symmetrical about the center line, while the core layer of the IMD/MIM is offset by 0.08 mm toward the film side.

The temperature field and viscosity field affect the boundary position of the core layer. As shown in Figure 14e,f, the highest point of the temperature curve and the lowest point of the viscosity curve of MIM are on the center line, while the highest point of the temperature curve and the lowest point of the viscosity curve of IMD/MIM are off the center line, approaching the film side, so the core layer of the IMD/MIM is offset toward the film side. The offset of the core layer affects the thickness of the transition layer. It can be seen from Figure 14c,d that the thicknesses of the two transition layers of MIM sample are similar. However, the thickness of transition layer B (with film side) of IMD/MIM sample is less than that of the transition layer A. The intersections of the four yellow dashed lines and the purple dashed line shown in Figure 14e,f represent the intersections of the above four boundary positions with the temperature profile at the end of the melt filling. It can be seen that the temperatures at these locations are almost all about 212 °C. Only the temperature at the left boundary of the IMD/MIM core layer in Figure 14f is slightly above 212 °C. The melt viscosity at this boundary is large, resulting in low melt strength, so cells easily collapse and merge, thus a higher temperature is required.

#### 4.3.2. Cellular Structure of Parallel Section

SEM micrographs of the parallel section of MIM and IMD/MIM samples are given in Figure 16. It can be seen that whether it is the MIM or IMD/MIM sample, the core layer cells are spherical, while the transition layer cells are severely deformed and appear elliptical. When the transition layer cells grow during the melt filling stage, they are stretched and deformed by the strong shearing action of the melt. Higher temperature of the core layer results in lower melt viscosity, which means the shear stress subjected is smaller, so most of the core layer cells are still in a spherical state. The cell density of the core layer is lower than that of the transition layer. Higher core layer temperature provides longer growth time for cells, while lower melt strength results in reduced cell growth resistance. Therefore, cells are more likely to merge or collapse, resulting in the cell density of the core layer being lower than that of the transition layer.

Due to the severe deformation of the transition layer cells, we no longer study the cell size, but use the ratio of the length to diameter to characterize the cell deformation and tilt angle to characterize the cell orientation. As can be seen from Figure 17a, it is not difficult to find that the ratio of the length to diameter and the tilt angle of the two transition layer cells of MIM are both similar. The length to diameter ratio of the transition layer B (film side) cells of the IMD/MIM is smaller than that of the transition layer A, and the tilt angle is larger than that of the transition layer A. This is because the temperature of the film side is higher, and the melt viscosity is lower, so the cells are less sheared during the melt filling stage, thus resulting in a smaller aspect ratio and smaller angle of the cell from the vertical direction, so the tilt angle is larger. The cell density on the film side is similar to that on the non-film side, which is consistent with the cell distribution phenomenon in the vertical direction.

From Figure 16a,b, we can see that the distance from the left edge of the MIM core layer to the left surface is 1.41 mm, and the distance from the right edge to the right surface is 1.39 mm, while the distance from the left edge of the IMD/MIM core layer to the left surface is 1.49 mm and the distance from the right edge to the right surface is 1.31 mm. From these data, we can find that the thickness of IMD/MIM core layer is the same as that of MIM, which is almost 1.2 mm, but the core layer of MIM is symmetrical about the center line, while the core layer of IMD/MIM is offset to the film side by 0.08 mm. Due to the reduced heat transfer coefficient on the film side, the highest temperature no longer appears on the center line of the thickness direction, but is offset toward the film side by a certain distance; therefore, the core layer of the IMD/MIM is biased toward the film side.

## 5. Conclusions

A combined in-mold decoration and microcellular injection molding (IMD/MIM) method was presented in this paper. The in-mold decoration injection molding (IMD), conventional injection molding (CIM), IMD/MIM and microcellular injection molding (MIM) simulations and experimental comparisons were performed to validate the effectiveness of the IMD/MIM method. The mechanical properties, forming defects and cellular structure of the samples were analyzed and compared.

The results show that the proposed IMD/MIM method can improve surface quality while maintaining almost no degraded mechanical properties and obtaining a cell-offset part. Compared with IMD, the mechanical properties of IMD/MIM samples are reduced due to the presence of cells. However, the specific mechanical properties are rarely reduced, and specific mechanical properties similar to their solid counterparts are maintained. Compared with MIM, the presence of the film flattens the bubbles that have not been cooled and turned to the surface, thus improving the surface quality by eliminating the bubble marks, and the IMD/MIM parts have almost the same good surface appearance as that of CIM. However, the existence of the film reduces the heat transfer coefficient on the film side, so that the two sides of the part are cooled asymmetrically, resulting in an asymmetry temperature, thus causing warpage. The presence of the film results in an asymmetrical temperature distribution along the thickness of the specimen. The higher temperature on the film side leads the cells to move toward it, obtaining a cell-offset part. Therefore, the method provides an effective way of producing foamed parts with improved surface appearance, which provides a broad development prospect for its applications in many industries such as furniture packaging, construction and automotive interior and exterior.

## Figures and Tables

**Figure 1 polymers-11-00778-f001:**
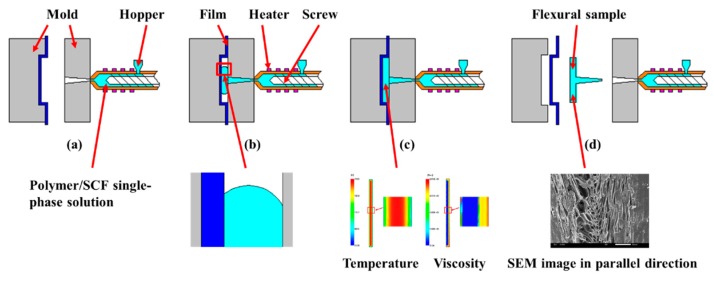
Schematic diagram of the IMD/MIM: (**a**) pasting film before injection; (**b**) injecting polymer/SCF single-phase solution; (**c**) cooling molding; (**d**) ejecting flexural sample.

**Figure 2 polymers-11-00778-f002:**
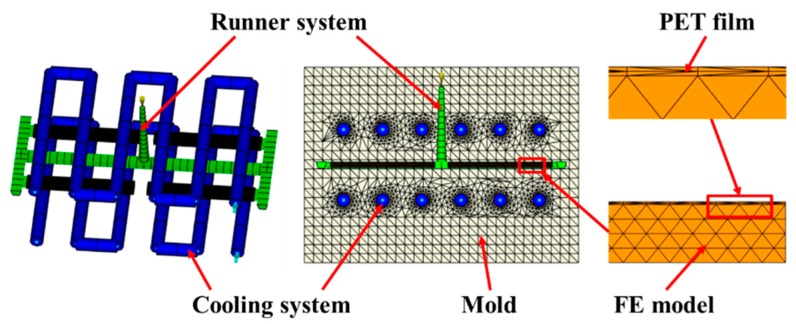
Finite element model.

**Figure 3 polymers-11-00778-f003:**
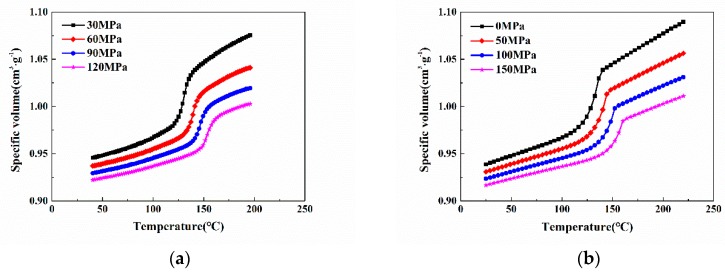
PVT performance of modified PP: (**a**) PVT curve of modified PP; (**b**) fitted PVT curve of modified PP.

**Figure 4 polymers-11-00778-f004:**
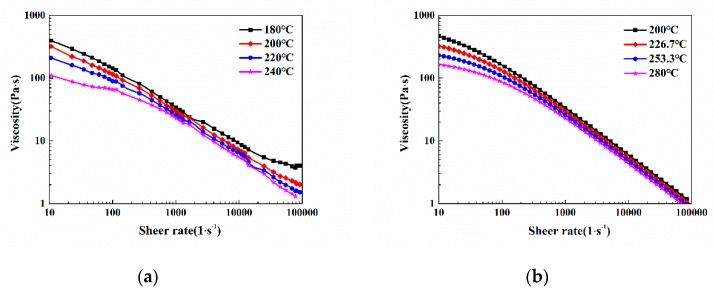
Viscosity property of modified PP: (**a**) viscosity curve of modified PP; (**b**) fitted viscosity curve of modified PP.

**Figure 5 polymers-11-00778-f005:**
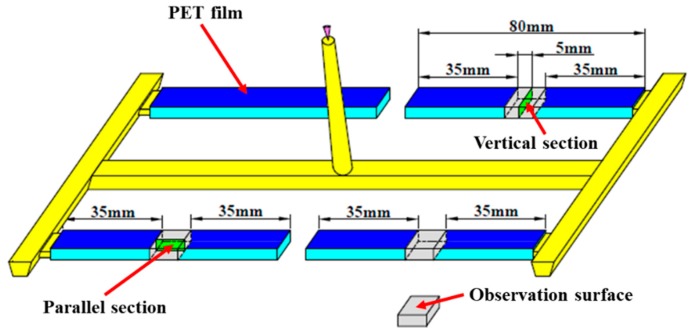
Preparation of samples for SEM observation from flexural samples.

**Figure 6 polymers-11-00778-f006:**
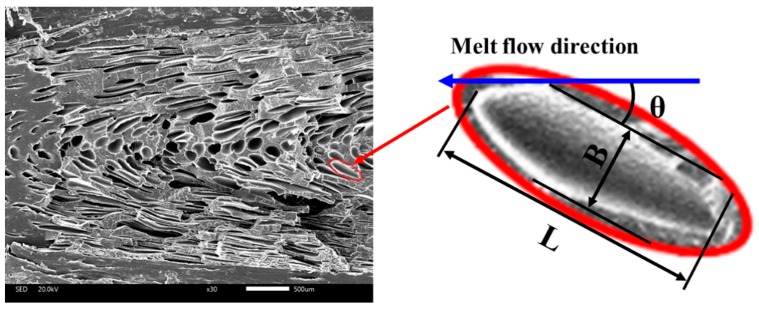
Schematic diagram of ratio of length to diameter and tilt angle.

**Figure 7 polymers-11-00778-f007:**
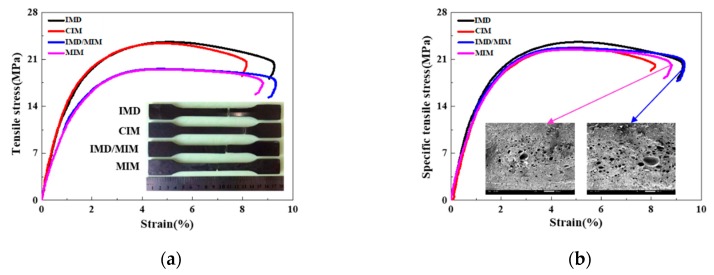
The tensile properties of the IMD, CIM, IMD/MIM and MIM samples: (**a**) tensile stress-strain curves; (**b**) specific tensile stress-strain curves.

**Figure 8 polymers-11-00778-f008:**
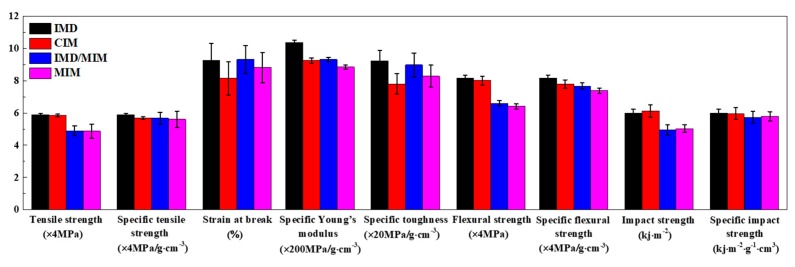
Comparison of mechanical properties among the IMD, CIM, IMD/MIM and MIM samples (the error bars in the figure represent the standard deviation).

**Figure 9 polymers-11-00778-f009:**
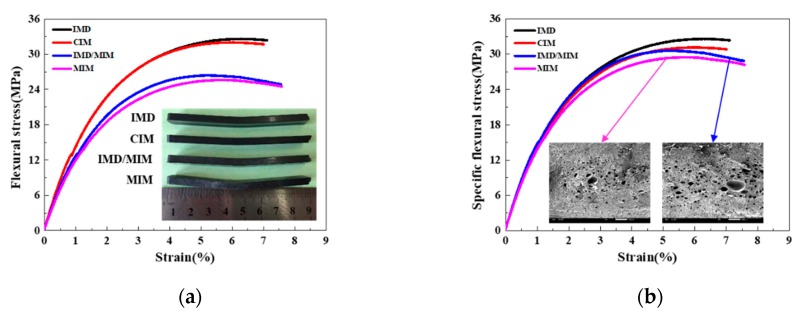
The flexural properties of the IMD, CIM, IMD/MIM and MIM samples: (**a**) flexural stress-strain curves; (**b**) specific flexural stress-strain curves.

**Figure 10 polymers-11-00778-f010:**
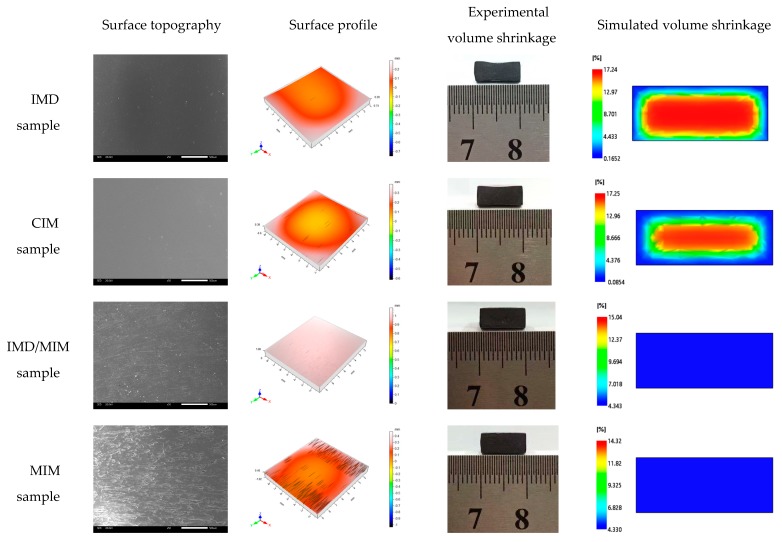
The comparison of the surface topography, surface profile, experimental volume shrinkage and simulated volume shrinkage among the IMD, CIM, IMD/MIM and MIM samples.

**Figure 11 polymers-11-00778-f011:**
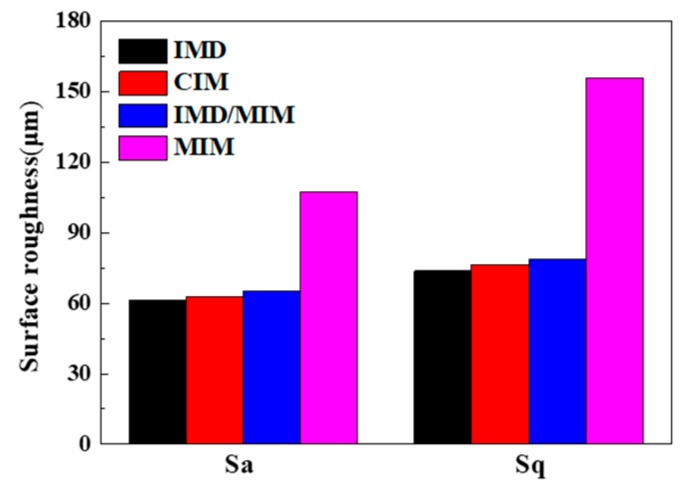
The comparison of the surface roughness indexes among the IMD, CIM, IMD/MIM and MIM samples.

**Figure 12 polymers-11-00778-f012:**
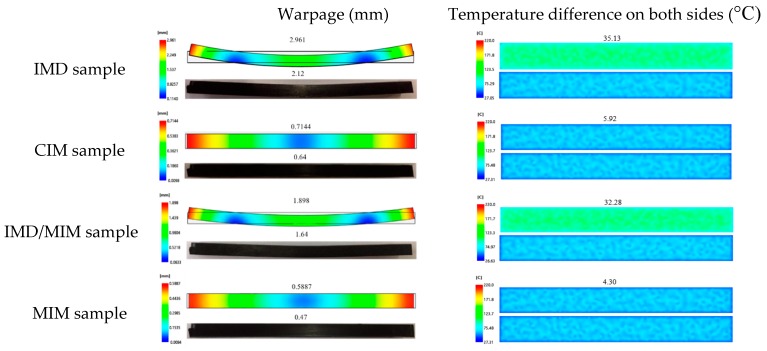
The comparison of the warpage and temperature difference on both sides among the IMD, CIM, IMD/MIM and MIM samples.

**Figure 13 polymers-11-00778-f013:**
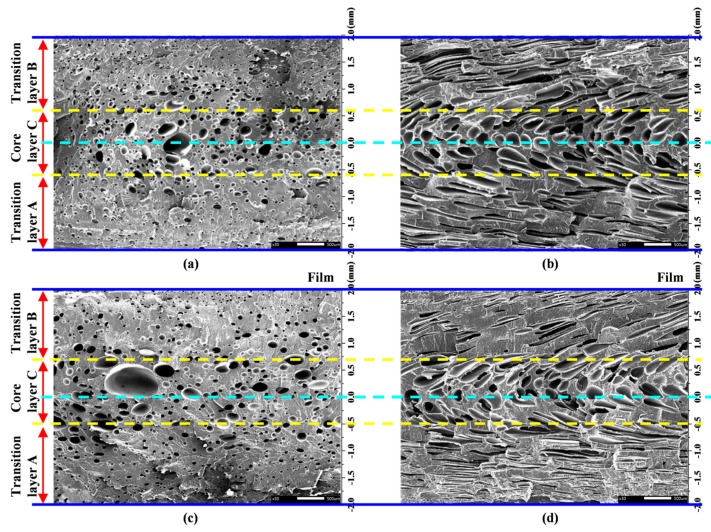
Layered schematic diagram: (**a**) and (**b**) layered schematic diagram of vertical section and parallel section of the MIM sample, respectively; (**c**) and (**d**) layered schematic diagram of vertical section and parallel section of the IMD/MIM sample, respectively.

**Figure 14 polymers-11-00778-f014:**
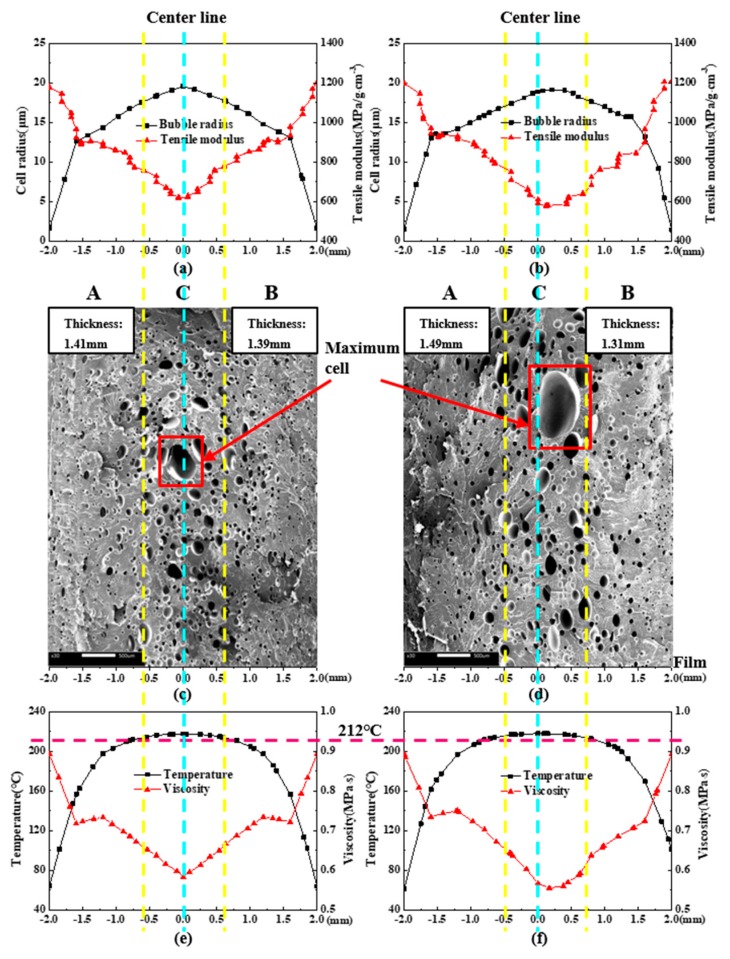
Simulation results and SEM micrographs of the vertical section of MIM and IMD/MIM samples: (**a**), (**b**) and (**e**), (**f**) the simulation results along the center line of vertical section of MIM and IMD/MIM samples, respectively; (**c**) and (**d**) the SEM micrograph of MIM and IMD/MIM samples, respectively.

**Figure 15 polymers-11-00778-f015:**
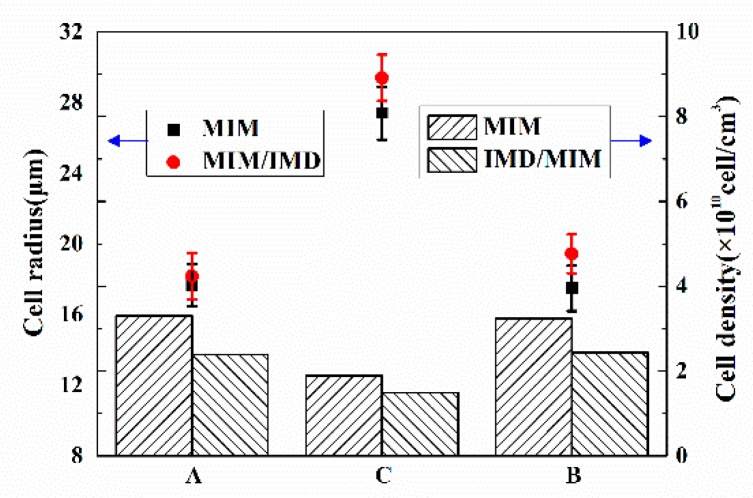
The cell radius and cell density of the transition layer A, core layer C and transition layer B of MIM and IMD/MIM samples.

**Figure 16 polymers-11-00778-f016:**
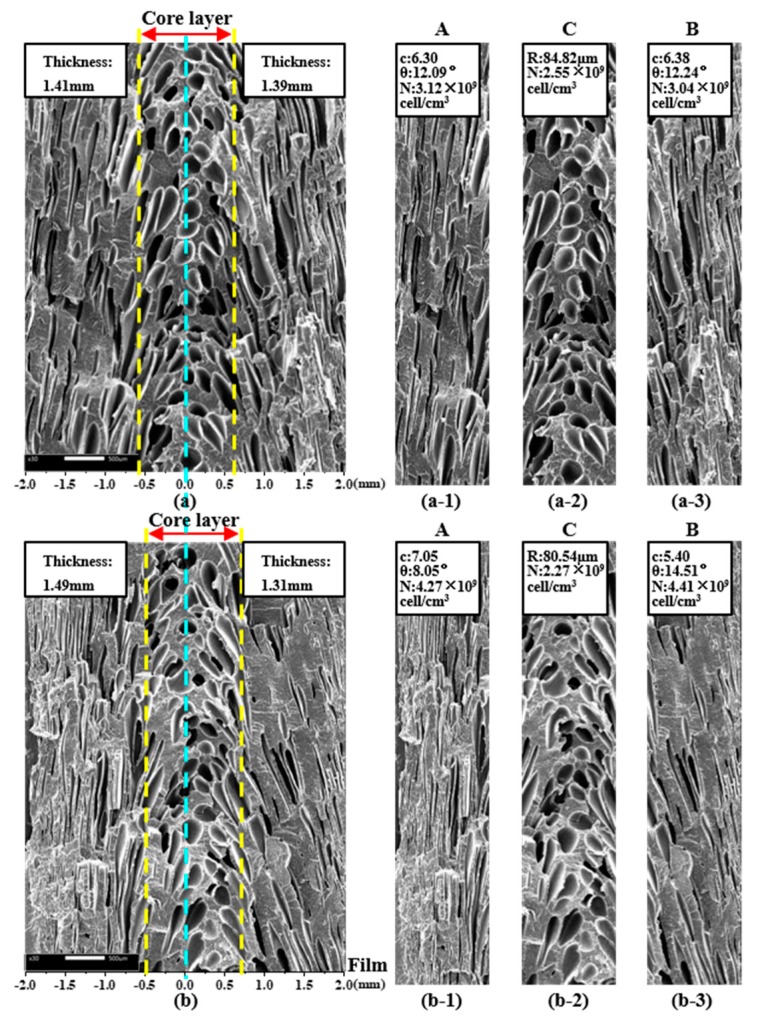
SEM micrographs of the parallel section of MIM and IMD/MIM samples: (**a**) and (**b**) the SEM micrograph of the parallel section of MIM and IMD/MIM samples, respectively; (**a-1**), (**a-2**), (**a-3**) and (**b-1**), (**b-2**), (**b-3**) the SEM micrographs of the transition layer A, core layer C and transition layer B of the parallel section of MIM and IMD/MIM samples, respectively.

**Figure 17 polymers-11-00778-f017:**
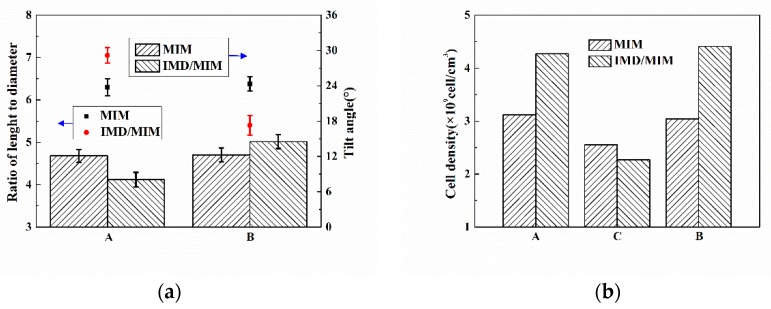
(**a**) The ratio of length to diameter and tilt angle of the transition layer A and transition layer B of MIM and IMD/MIM samples; (**b**) The cell density of the transition layer A, core layer C and transition layer B of MIM and IMD/MIM samples.

**Table 1 polymers-11-00778-t001:** Physical properties of the materials.

Family Abbreviation	Modified PP	PET
Brand	AIP-1927	543C
Solid density (g/cm^3^)	1.0278	1.4050
Melt density (g/cm^3^)	0.8682	1.1696
Elastic modulus (MPa)	1300	3450
Sheer modulus (MPa)	780	1225
Poisson’s ratio	0.350	0.408

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
