# Peer review of "A Combined In-Mold Decoration and Microcellular Injection Molding Method for Preparing Foamed Products with Improved Surface Appearance"

_polymers, 2019, doi:10.3390/polym11050778_

Round 1

Reviewer 1 Report

In the revision of the article authors have adressed the requested points. Therefore in my opinion the manuscript is now suitable for publication.

Reviewer 2 Report

Figure 10 now fig. 9 is still the same, please update picture according to reviewer comments.

Fig. 11(Fig. 10 now) has not been modified, please update picture according to reviewer comment

This manuscript is a resubmission of an earlier submission. The following is a list of the peer review reports and author responses from that submission.

Round 1

Reviewer 1 Report

In the opinion of the reviewer:

-        It would be recommendable that the authors carefully proof read their paper

-        Sentence from line 201 – 204 and 221 – 224 are basically the same, please rephrase.

-        Figure 10 scale bar indicate what? Please state this in the figure caption.

-        In surface topography section:

o   Figure 11 should be completely revisited considering that none of the images are visible as well as the scale bars

o   Line 248 is not visible, what does it means smooth? Compared to what? This is not a scientific description of your surface topography. Please rephrase the sentence.

o   Looking at figure 12: What instrument has been used for the characterization? What is the estimated uncertainty of your measurements? How many measurement repetition have you performed? How many sample were measured?

o   Line 258 surface profile: this is not visible as well the dimensional scale

-        Figure 15: nice impression but difficult to be interpreted please consider to simplify it, yellow notes on the SEM pictures are not readable  

-        Figure 17: yellow notes on the SEM pictures are not readable

-        Finally It would be recommendable that after your concluding sentence (line 426-427) you list potential industrial or product applications that would benefit from the improved foamed surface appearance

Reviewer 2 Report

The present manuscript “A combined in-mold decoration and microcellular injection molding method for preparing foamed products with improved surface appearance” is about the enhancement of the surface properties of injection molding samples made through the MuCell technology for foaming. 
The most important novelty of the work is the combination of two existing techniques, in mold decoration and Mucell, for improving the external aspect of the obtained surfaces. It is worth to mention that the combination of different technologies for avoiding external defects (Rapid heat-cool process, co-injection molding, gas counter-pressure etc.) has been done in the past, and the combination of in-mold decoration with Mucell is a logical continuation. However, there are not previous studies in literature dealing with this system which is in fact an important novelty of the present research 
Nevertheless, in my opinion the novelty of the paper alone, is not enough to justify the publication in polymers. Some of the results obtained are either not well explained or not well supported by data. Please see the comments below for details. The standard of English is another problem. I have found many errors both in the construction of sentences and in the spelling. So I recommend a complete revision of the paper language. The English revision of the paper has not been included in the comments.
